# Long-Term Analysis of Wave Climate and Shoreline Change along the Gulf of California

**Cuauhtémoc Franco-Ochoa** [1,*] **, Yedid Zambrano-Medina** [2] **, Wenseslao Plata-Rocha** [2] **,**
**Sergio Monjardín-Armenta** [2] **, Yandy Rodríguez-Cueto** [3] **, Mireille Escudero** [3]
**and Edgar Mendoza** [3]

[1] Department of Civil Engineering, Autonomous University of Sinaloa, Culiacán Rosales 80040, Mexico

[2] Department of Earth and Space Sciences, Autonomous University of Sinaloa,
Culiacán Rosales 80040, Mexico; yedidzambrano@uas.edu.mx (Y.Z.-M.);
wenses@uas.edu.mx (W.P.-R.); sa.monjardin12@info.uas.edu.mx (S.M.-A.)

[3] Engineering Institute, National Autonomous University of Mexico, Mexico City 04510, Mexico;
yandyro84@gmail.com (Y.R.-C.); mescuderoc@iingen.unam.mx (M.E.); emendozab@iingen.unam.mx (E.M.)

\* Correspondence: cfrancoo@uas.edu.mx

**Abstract:** The last ten years have shown that Climate Change (CC) is a major global issue to attend to. The integration of its effects into coastal impact assessments and adaptation plans has gained great attention and interest, focused on avoiding or minimizing human lives and asset losses. Future scenarios of mean sea level rises and wave energy increase rates have then been computed, but downscaling still remains necessary to assess the possible local effects in small areas. In this context, the effects of CC on the wave climate in the Gulf of California (GC), Mexico, have received little attention, and no previous studies have tackled the long-term trend of wave climate at a regional scale. In this paper, the long-term trends of the wave height, wave period and wave energy in the GC were thus investigated, using the fifth-generation climate reanalysis dataset (ERA5). The long-term shoreline evolution was also examined from historical Landsat images, so as to identify erosional hotspots where intervention can be prioritized. The results indicate that both the mean and extreme wave regimes in the GC are getting more energetic and that two-thirds of the coast is suffering chronic erosion. A discrepancy between the trends of the wave period and wave height in some regions of the Gulf was also found. Finally, the importance of natural processes, human activity and CC in the shoreline change is highlighted, while addressing the need for future permanent field observations and studies in the GC.

**Keywords:** shoreline change; wave climate; climate change; Gulf of California

## 1. Introduction

Climate Change (CC) is the greatest environmental threat faced by humanity. The effects of CC will be particularly noticeable on the coast, where humans and oceans meet [1]. In the coastal environment, the effects of CC lead to variations in river flows and ocean/atmosphere interactions, leading to some sea level rises, increased wind-generated waves and storm surges [2], which can cause extreme coastal erosion and flooding [3].

Currently, the global mean sea level and wave energy are increasing at rates of 3.26 mm [4] and 0.4% [1] per year, respectively. However, a precise quantification of regional rates is much harder and depends on factors such as the local wind energy, the sea water temperature and tectonic uplift/subsidence motions, among others [1,4].

In addition, it is estimated that 24 to 70% of the sandy beaches around the world are being eroded [5–7], and considering the projected variations in sea levels and wave energy, the erosion rates and the percentage of threatened sandy beaches are likely to increase. This highlights the relevance of integrating the effects of CC into coastal impact assessments, in order to generate more accurate adaptation plans.

According to [8], the recent sea level rise (1973–2015) in the Gulf of California (GC), Mexico, has a geographical gradient, i.e., the upper Gulf presents a higher mean trend, while it is lower in the middle part of the Gulf. In the lower Gulf, the mean trend shows a transition from the entrance of the Gulf to the middle Gulf (see Figure 1). Quantitatively, the regional mean sea level trend, estimated on one side with satellite altimetry data and on the other side with measurements from tide gauges located inside the Gulf, is 0.8 ± 0.8 mm and 2.5 ± 1.1 mm per year, respectively. Nevertheless, the possible inaccuracy of these sea level trend values, obtained from both methods, and the observed discrepancies highlight the need for further efforts to corroborate the results and better understand the non-uniform sea level change across the GC [9].

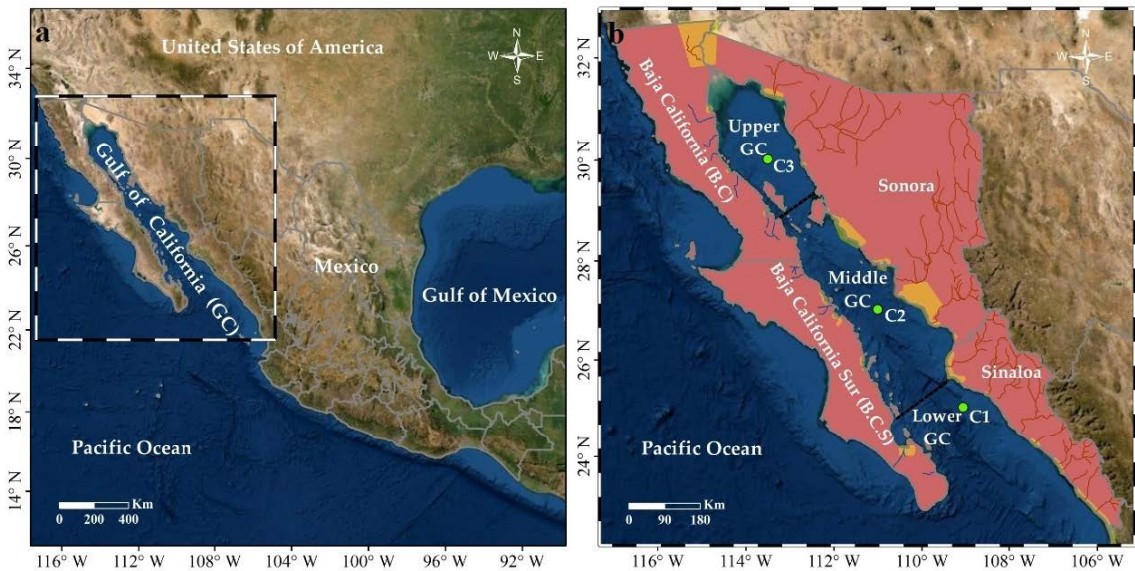

**Figure 1.** Study area: (**a**) macro-location; (**b**) micro-location, indicating the high population density, as well as the main economic activities including tourism, agriculture, aquaculture and all sorts of fisheries (yellow shading), the river network (undammed river in blue and dammed river in red), and the location of the three wave data sources (C1, C2 and C3).

In general, wave climate has received less attention in the GC, particularly in the context of climate change. Previous studies (e.g., [10–12]), using wave data via the reanalysis of different databases, have focused on analyzing the seasonal to long-term variability of the directional wave power at a regional scale, while defining the mean and extreme conditions of wave climate at a local scale. However, none of these studies have covered the long-term trend of wave climate at a regional scale, despite its significance for the development of efficient plans for adaptation to CC [12]. Moreover, global-scale studies (e.g., [1,13]) on the long-term trend in wave energy and wave height do not cover the GC, or they do it only partially. This knowledge gap makes difficult the integration of the effects of CC into regional- and local-scale coastal impact assessments, required for adaptation plans. To address this, the present work investigated whether or not there was a trend in the wave period, wave height and wave power, using the ERA5 dataset [14], which provides a 40-year (1980–2019) time series of wave data. Furthermore, given the importance of knowing the shoreline evolution in the design of those CC adaptation plans and priorities [15–18], a long-term analysis of the shoreline change all along the GC coast was performed using the Digital Shoreline Analysis System (DSAS) (version 5.0) [19].

The paper is divided into six sections. Following the introduction in Section 1, Section 2 presents the study area, and Section 3, the materials and methods. The results for the trends of wave climate and long-term shoreline change all along the Gulf are described in Section 4. Section 5 discusses the findings described in Section 4. The conclusions of the study are presented in Section 6.

## 2. Study Area

The study area is the Gulf of California, also known as the Sea of Cortés or Sea of Bermejo. It is located in the northwestern part of Mexico (Figure 1a) and is bordered by the states of Sonora and Sinaloa in mainland northwest Mexico, as well as the Baja California Peninsula (Figure 1b). It is one of the most productive and diverse marine ecosystems in the world [20]. Currently, it is mostly in a relatively good state of conservation, but it has started to be threatened by increasing human pressure [21] (see Figure 1b). The length of its west coast (Baja California Peninsula) is 1640 km, and the length of its east coast (Sonora and Sinaloa states) is estimated to be 1731 km; the mean cross-shore distance from the west to east coast varies from 110 to 220 km.

Due to its geographical location (within the transition zone between tropical and subtropical climate regimes), the GC is exposed to multi-year processes, such as the El Niño–Southern Oscillation (ENSO) and the Pacific Decadal Oscillation (PDO) [22], and other phenomena such as the North American Monsoon (NAM) (also known as the Southwest Monsoon) and cold fronts. The Southwest Monsoon usually covers the period from July to September and is related to the onset of rainfall, tropical cyclones (hurricanes and depressions), the formation of a low-pressure center and the change of northwesterly winter winds to winds from the south or even the southeast [23–26]. The cold fronts last from September to March, with northwesterly winds blowing over the GC [27,28].

On the west coast, the beaches are characterized by coarse sands (0.50–1.00 mm), and the mean beach slope is 7°. The river network is non-perennial, so the contribution of new sediment to these beaches only occurs after storms or heavy rains. On the east coast, the beach sands are mainly medium in size (0.25–0.50 mm), and the mean beach slope is 4°. The river network there is almost perennial, but most of the rivers are dammed, which can reduce the transport of sediment to the coast during the dry season. Rivers that are not dammed have some small impoundments for irrigation but still largely freely run towards the sea [29]. An important aspect of the coastal profile of the GC is that the beach and continental shelf on the west coast are narrower than the ones on the east coast, and in some stretches (mainly in the state of Baja California Sur), the beach is absent [30]. Figure 2 shows some images of the GC coast.

The astronomical tide is predominantly of a mixed type, with trends towards the semidiurnal type in the northern upper Gulf and towards the diurnal type in the middle Gulf, especially on the east coast [31]. The mean tidal range shows an upward trend from the lower to the upper Gulf, ranging from less than 1.0 m in the lower Gulf up to approximately 3.5 m in the upper Gulf [32]. Regarding the waves, these are of low height (~0.30 m) in the upper Gulf, while in the middle Gulf, 0.60 to 0.90 m waves are found, and in the lower Gulf, heights from 1.50 to 1.80 m have been reported [29].

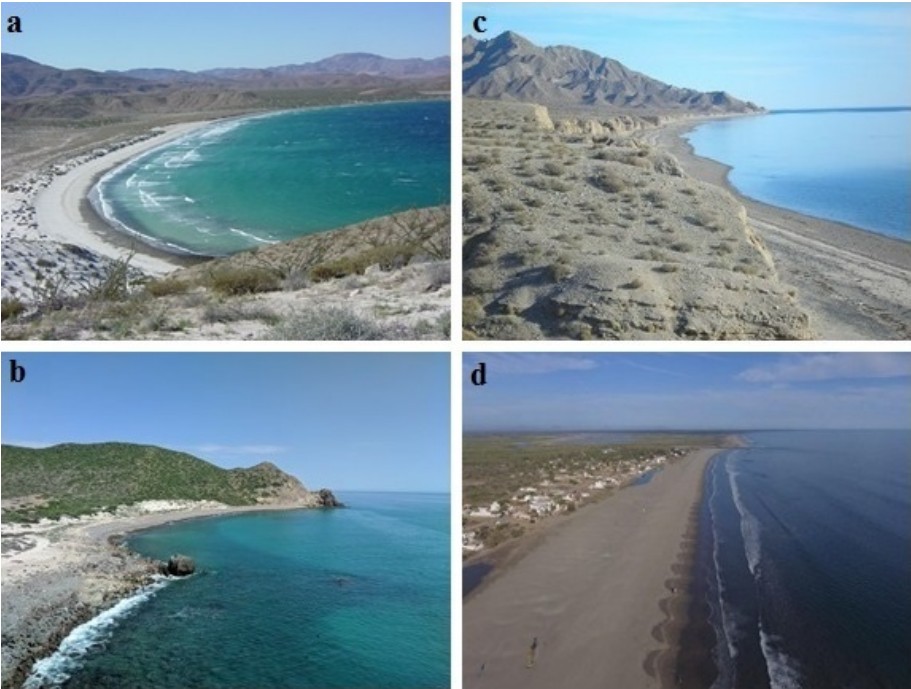

**Figure 2.** Images of the west and east coasts of the Gulf of California (GC): (**a**) Baja California (B.C.); (**b**) Baja California Sur (B.C.S.); (**c**) Sonora; (**d**) Sinaloa. Source: Google Earth (2018).

## 3. Materials and Methods

### 3.1. Shoreline Data

In this paper, historical Landsat images from the Earth Explorer database of the United States Geological Survey (USGS) (available at http://www.glovis.usgs.gov), which cover the whole coast of GC, were used to detect shoreline changes between 2019 and the time of the oldest available Landsat image (between 1978 and 1981). All the images were collected in autumn in order to reduce the effects of seasonal variations of the mean sea level and wave energy. Only images with similar quality and cloudiness were used. The characteristics of the satellite images used to examine the shoreline position are listed in Table 1.

### 3.2. Image Processing

Two major image-processing operations were carried out in this study. First, radiometric and atmospheric correction was applied to all the satellite images of Table 1 in the ENVI 5.3 software [33]. The Reflectance Radiometric Calibration method was used to sharpen the satellite images, while the Fast Line of Sight Atmospheric Analysis of Spectral Hypercubes (FLAASH) method eliminated disturbances caused by the atmosphere [34]. After that, the Normalized Difference Water Index (NDWI) [35] was calculated with bands 4 (Green) and 6 (Near InfraRed (NIR)) of the Multispectral Scanner System (MSS) sensor, and bands 3 (Green) and 5 (NIR) of the Operational Land Imager (OLI) sensor, through Equation (1):

$$\text{NDWI} = \frac{\text{Green} - \text{NIR}}{\text{Green} + \text{NIR}} \tag{1}$$

The relation between the green and NIR bands highlights in the images the conditions of high soil moisture and water masses [35–37]. In turn, it highlights the upper wetting limit of the foreshore caused by the highest run-up of the last high tide before the satellite image acquisition date.



**Table 1.** Historical Landsat images used in the shoreline position analysis.

| Spacecraft ID/Sensor | Acquired Date | Path/Row | Pixel Resolution |
|---|---|---|---|
| Landsat 2-3/Multispectral Scanner System (MSS) | 1978-11-05 | 036/043 | 60 m |
| | 1978-11-05 | 036/044 | 60 m |
| | 1978-10-01 | 037/042 | 60 m |
| | 1978-10-01 | 037/043 | 60 m |
| | 1978-10-02 | 038/041 | 60 m |
| | 1978-10-02 | 038/042 | 60 m |
| | 1978-11-08 | 039/040 | 60 m |
| | 1978-11-08 | 039/041 | 60 m |
| | 1980-10-19 | 039/040 | 60 m |
| | 1980-10-02 | 040/039 | 60 m |
| | 1981-10-26 | 033/044 | 60 m |
| | 1981-10-27 | 034/043 | 60 m |
| | 1981-11-15 | 035/043 | 60 m |
| | 1981-11-15 | 035/042 | 60 m |
| | 1981-11-16 | 036/042 | 60 m |
| | 1981-09-23 | 036/041 | 60 m |
| | 1981-09-23 | 036/042 | 60 m |
| | 1981-11-17 | 037/041 | 60 m |
| | 1981-10-14 | 039/039 | 60 m |
| | 1981-09-27 | 040/038 | 60 m |
| Landsat 8/Operational Land Imager (OLI) | 2019-10-31 | 031/044 | 30 m |
| | 2019-10-22 | 032/043 | 30 m |
| | 2019-10-22 | 032/044 | 30 m |
| | 2019-10-29 | 033/042 | 30 m |
| | 2019-10-29 | 033/043 | 30 m |
| | 2019-10-04 | 034/042 | 30 m |
| | 2019-10-20 | 034/044 | 30 m |
| | 2019-10-11 | 035/041 | 30 m |
| | 2019-10-11 | 035/042 | 30 m |
| | 2019-11-14 | 033/044 | 30 m |
| | 2019-10-20 | 034/041 | 30 m |
| | 2019-10-16 | 038/039 | 30 m |
| | 2019-10-04 | 034/043 | 30 m |
| | 2019-10-16 | 038/038 | 30 m |
| | 2019-10-25 | 037/040 | 30 m |
| | 2019-10-25 | 037/039 | 30 m |
| | 2019-10-25 | 037/038 | 30 m |
| | 2019-10-18 | 036/041 | 30 m |
| | 2019-10-18 | 036/040 | 30 m |

*3.3. Extraction and Shoreline Analysis*

The High Water Line (HWL) is widely used in the investigation of shoreline change to represent the shoreline [38]. It is defined as the highest run-up of the last high tide [39] that is visually discernible in satellite images [16]. Its demarcation in the images is difficult due to the lack of knowledge of the oceanographic conditions at the time of the satellite image acquisition. Nevertheless, it has been shown that the wet/dry line (on a rising tide, it is equal to the maximum run-up limit; on a falling tide, it is equal to the part of the beach that is still wet [16]) closely approximates the HWL [40,41]. For these reasons, in this paper, the wet/dry line from the Landsat images was used as an approximation of the HWL inland/outland horizontal displacement. It was manually digitized from each satellite image in ArcGIS 10.5. Given the latter and the resolution of the images, the objective of the analysis was to evaluate the erosion and accretion long trends and not the precise quantification of the shoreline displacements based on the position of the HWL.

Shoreline changes were analyzed using DSAS (version 5.0), which is a software extension for ArcGIS©, developed by the USGS to calculate shoreline change rates from multiple historic shoreline positions [19]. To use DSAS, a geographic database was created that combined digitized shorelines and a baseline in front of the coast, then cross-shore transects every 100 m were generated along the west and east coasts of the GC. End Point Rate (EPR) statistics were calculated for each transect (the EPR being the division of the distance between the shoreline position at different dates by the difference in years of the shoreline position) using the shoreline positions indicated in Figure 3 [42]. The EPR obtained for the GC coast was further divided into 4 categories (Table 2).

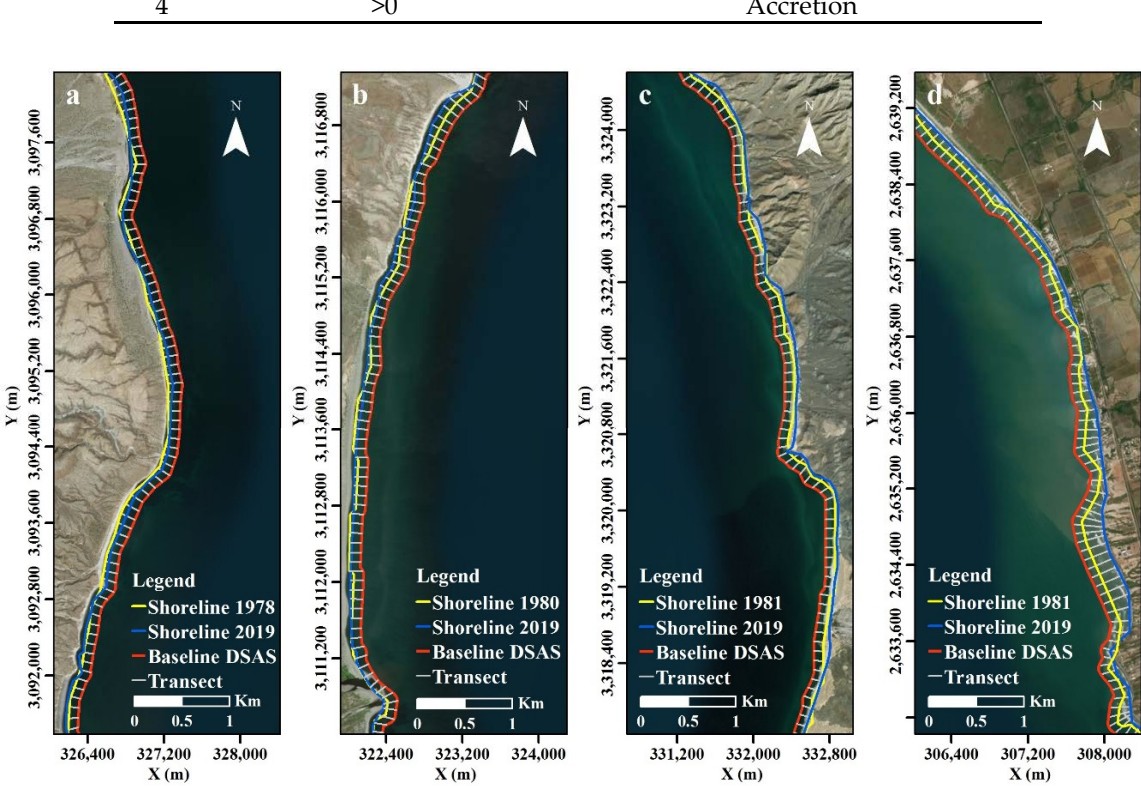

**Figure 3.** Examples of Digital Shoreline Analysis System (DSAS) transects and baseline map used for the analysis of the shoreline change along the study area: (**a**) B.C.S.; (**b**) B.C.; (**c**) Sonora; (**d**) Sinaloa.

**Table 2.** Shoreline classification based on End Point Rate (EPR).

| N° | EPR (m Per Year) | Shoreline Classification |
|----|------------------|--------------------------|
| 1 | $<-2$ | High erosion |
| 2 | $>-2$ to $<-1$ | Moderate erosion |
| 3 | $>-1$ to $<0$ | Low erosion |
| 4 | $>0$ | Accretion |

Finally, the shoreline change in the study area was evaluated for each state for two main reasons: (1) in order to facilitate the analysis and interpretation of the calculations, and (2) due to the availability of images for the study area, which were not homogeneous in time and space, and, thus, different periods of time were available for the analysis on each state (Figure 3).

*3.4. Offshore Wave Data*

Wave data were extracted from ERA5 dataset produced by the European Centre for Medium-Range Weather Forecasts (ECMWF) (available at http://www.ecmwf.int/research/era), which provides hourly

sea states from January 1980 to December 2019 (40 years) at three offshore wave reanalysis points: C1 (25° N, 109° W), C2 (27° N, 111° W) and C3 (30° N, 113.5° W) (Figure 1).

### 3.5. Wave Climate Analysis

The wave trend analysis was performed using the least-squares regression technique, applied to the long time series of wave data. This methodology was previously used by [43] to analyze the interdecadal variability of the wave characteristics on the west Guangdong coast in China. Additionally, the absolute wave power level (*P*) was calculated by Equation (2):

$$P = \frac{\rho g^2 H_s^2 T_s}{64\pi} \qquad (2)$$

where $\rho$ is the water density, $g$ is the gravity acceleration constant, $H_s$ is the significant wave height and $T_s$ is the mean wave period.

The time series of the annual means of $H_{99}$ (the 99th percentile), $H_s$, $T_s$ and $P$ were used to estimate the mean change rate per year (*R*) given by Equation (3):

$$R = \frac{\sum (x_i - \overline{x})(y_i - \overline{y})}{\sum (x_i - \overline{x})^2} \qquad (3)$$

where $x_i$ is the i-th year of the time series; $y_i$ is the i-th observation of the time series of the annual means of $H_{99}$, $H_s$, $T_s$ or $P$; and $\overline{x}$ and $\overline{y}$ are the means of $x_i$ and $y_i$, respectively.

The Mann–Kendall (MK) test [44,45] was used here to determine the significance of the wave data trends. This non-parametric test of randomness versus trend was previously used in other studies related to wave climates such as those by [1] and [46]. The purpose of this test was to statistically assess whether or not there was a consistent upward or downward trend in the variable of interest over time. The variance obtained from the MK statistic was then used to define the value for a specific significance level.

The multi-year means of $H_{99}$, $H_s$, $T_s$ and $P$ ($H_{99}$*, $H_s$*, $T_s$* and $P$*, respectively) were also calculated at the three locations (C1, C2 and C3), as the means of annual values, in order to obtain an overview of the spatial variation of these wave parameters along the GC.

## 4. Results

### 4.1. Long-Term Shoreline Change

The long-term shoreline analysis (Figure 4 and Table 3) showed that the major part of the coastline in the Gulf of California falls under the erosion category. The state with a major percentage of shoreline subject to erosion was Sonora (87.7% of its total shoreline), followed in descending order by Sinaloa (77.1% of its total shoreline), Baja California Sur (B.C.S.) (55.3% of its total shoreline) and Baja California (B.C.) (43.7% of its total shoreline). Erosion was mainly found to occur around the mouths of dammed rivers and in areas with high population densities, as well as agricultural and aquacultural land uses (Figure 4). The most prominent erosion hotspots were observed on the northern coast of B.C. (Figure 4a), on the northern and southern coasts of Sonora (Figure 4c), and along the coast of Sinaloa (Figure 4d). Accretion was mostly noticed along the coasts of B.C. and B.C.S. (except the north coast) (Figure 4a,b).

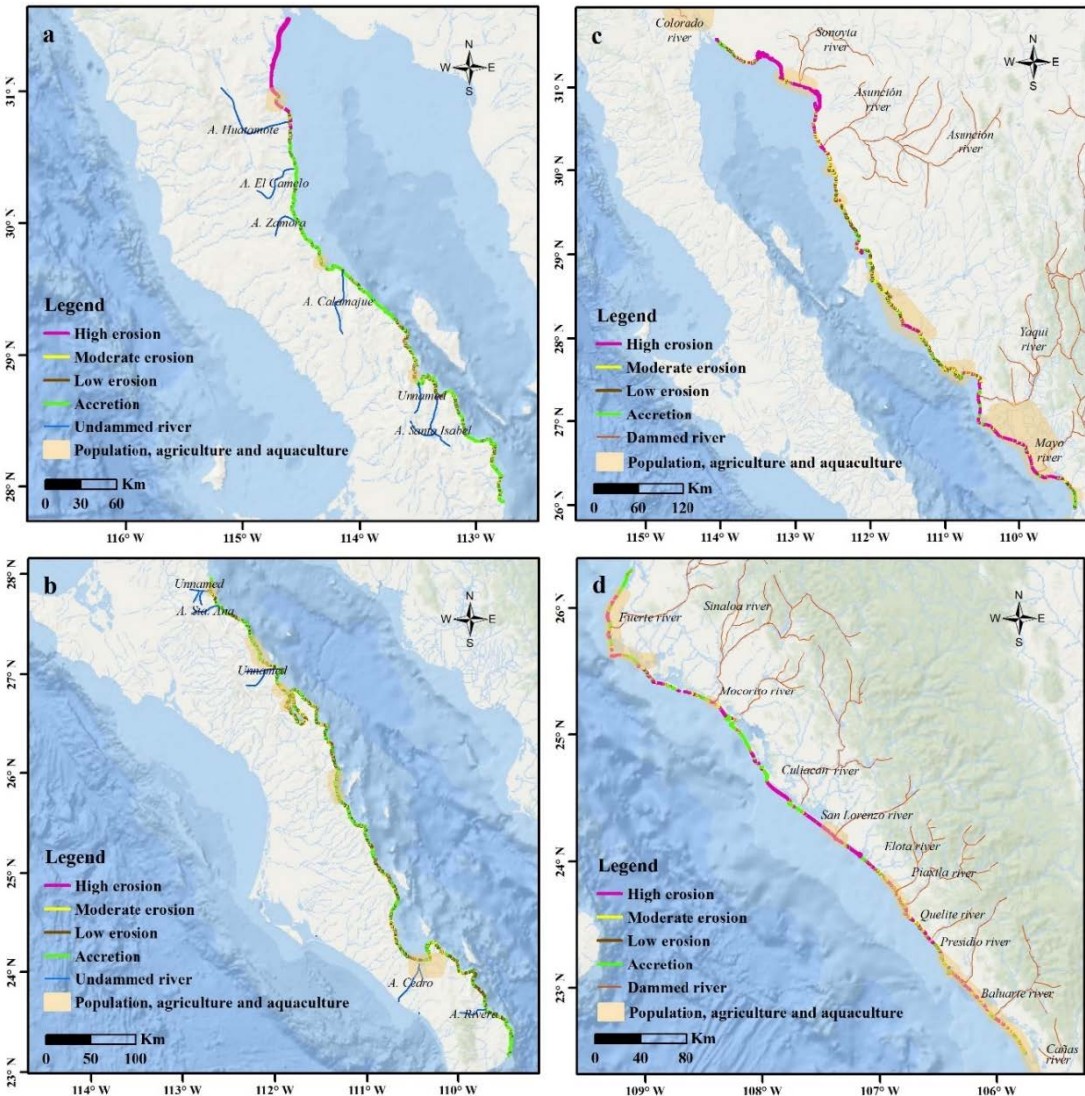

**Figure 4.** Long-term shoreline change maps: (**a**) B.C.; (**b**) B.C.S.; (**c**) Sonora; (**d**) Sinaloa.

**Table 3.** Long-term shoreline change statistics at state and regional scales.

| Statistics | B.C.S. | B.C. | Sonora | Sinaloa | Regional (Total) |
|---|---|---|---|---|---|
| Mean shoreline change (m/year) | −0.1 | −2.7 | −4.6 | −1.2 | −2.2 |
| Minimum shoreline change (m/year) | −4.8 | −43.4 | −78.1 | −50.7 | −78.1 |
| Maximum shoreline change (m/year) | 9.1 | 18.1 | 14.1 | 81.3 | 81.3 |
| Mean accretion rate (m/year) | 0.7 | 1.0 | 1.7 | 7.5 | 1.9 |
| Mean erosion rate (m/year) | −0.8 | −7.5 | −5.5 | −3.8 | −4.2 |
| Standard deviation (m/year) | 1.0 | 9.2 | 10.6 | 9.8 | 8.6 |
| Shoreline length that records erosion (km) | 558.2 | 275.0 | 958.7 | 491.7 | 2283.6 |
| Shoreline length that records accretion (km) | 451.7 | 354.9 | 134.4 | 146.4 | 1087.4 |
| Shoreline part subjected to erosion with respect to the total study area (%) | 16.6 | 8.2 | 28.4 | 14.6 | 67.8 |
| Shoreline part subjected to accretion with respect to the total study area (%) | 13.4 | 10.5 | 4.0 | 4.3 | 32.2 |

## 4.2. Trend and Multi-Year Mean of Wave Climate

The multi-year mean of $H_{99}$ showed the highest values at C2 ($H_{99}^* = 2.38$ m), followed in descending order by C3 ($H_{99}^* = 2.37$ m) and then by C1 ($H_{99}^* = 2.33$ m), although the difference between the highest and lowest values was less than 2.2% (Figure 5a). On the other side, the multi-year mean values of $H_s$, $T_s$ and $P$ at the three sites (Figure 5b–d) showed the highest parametric values at C1 ($H_s^* = 0.88$ m, $T_s^* = 7.78$ s and $P^* = 3.19$ kW/m), which gradually decreased to the north, with $H_s^* = 0.62$ m, $T_s^* = 4.49$ s and $P^* = 1.33$ kW/m at C2, and $H_s^* = 0.54$ m, $T_s^* = 3.32$ s and $P^* = 1.05$ kW/m at C3.

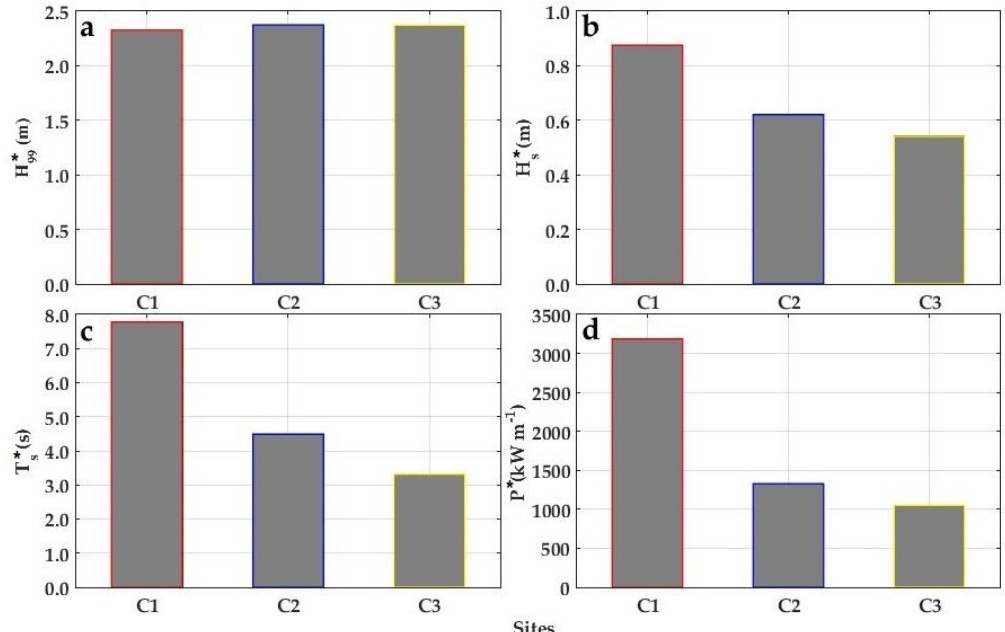

**Figure 5.** Multi-year means of $H_{99}$, $H_s$, $T_s$ and $P$ at C1, C2 and C3 for the whole 40-year period (1980–2019): (**a**) $H_{99}^*$; (**b**) $H_s^*$; (**c**) $T_s^*$; (**d**) $P^*$.

Furthermore, the annual mean of $H_{99}$ showed upward trends of 0.0099 m (0.99%), 0.0175 m (1.75%) and 0.0034 m (0.34%) at C1, C2 and C3, respectively (Figure 6a). Additionally, the annual mean of $H_s$ showed upward trends at C1, C2 and C3 of 0.0009 m (0.09%), 0.0022 m (0.22%) and 0.0010 m (0.10%) per year, respectively (Figure 6b). The annual mean of $T_s$ showed an upward trend at C3 of 0.0028 s (0.28%) per year, and downward trends at C1 and C2 of −0.0096 s (−0.96%) and −0.0010 s (−0.10%) per year, respectively (Figure 6c). Nonetheless, the annual mean of $P$ showed upward trends at these three sites: 0.0063 kW/m (0.63%) per year at C1, 0.0139 kW/m (1.39%) per year at C2 and 0.0045 kW/m (0.45%) per year at C3 (Figure 6d). The time series of the annual means of $H_{99}$, $H_s$, $T_s$ and $P$ at all the sites showed statistically significant trends at a 90% accuracy level.

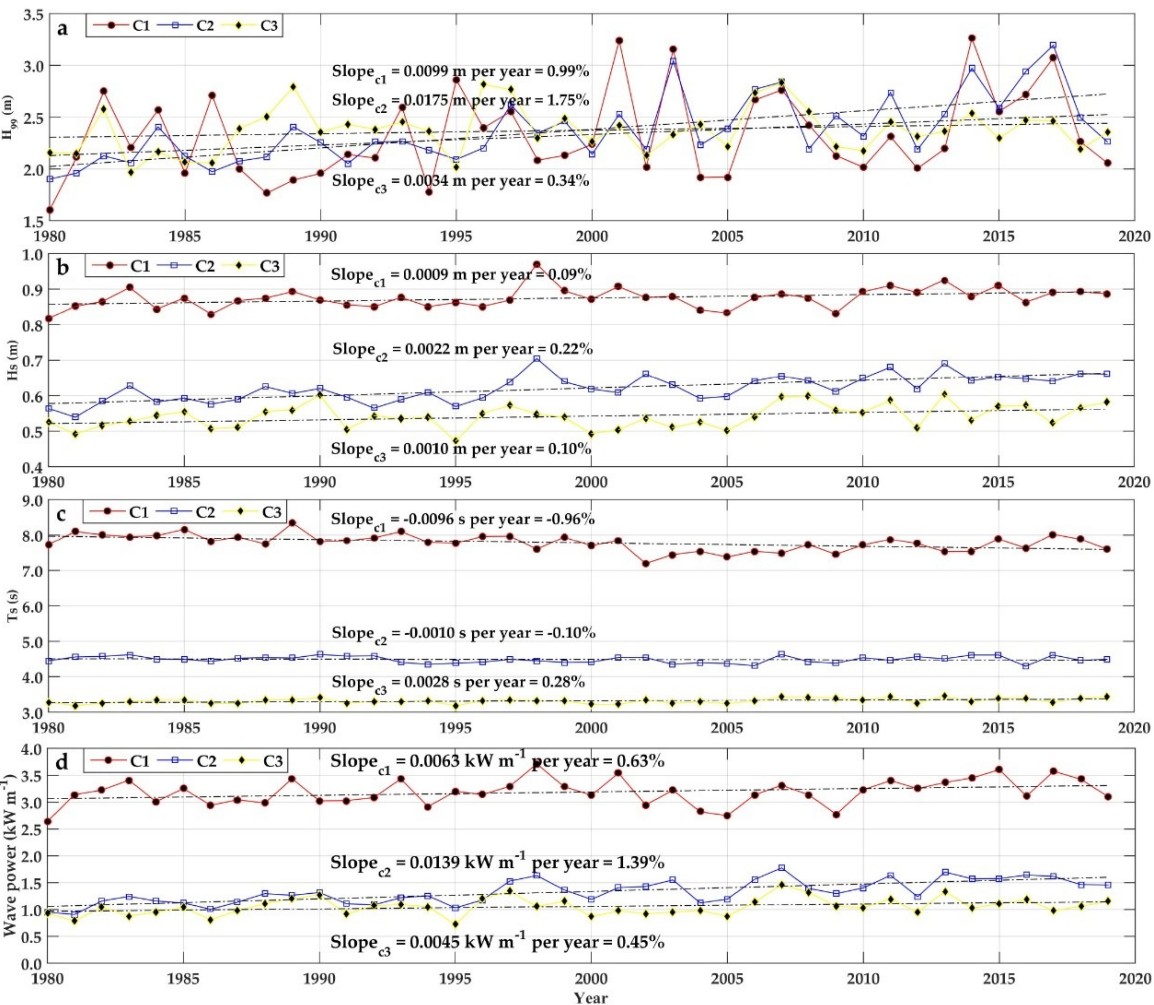

**Figure 6.** Variability of annual means of $H_{99}$, $H_s$, $T_s$ and $P$ at C1, C2 and C3 during the 40-year period (1980–2019): (**a**) $H_{99}$; (**b**) $H_s$; (**c**) $T_s$; (**d**) $P$.

## 5. Discussion

The results from the long-term wave climate analysis between 1980 and 2019 reveal a spatial variability in the multi-year mean values and long-term trends of $H_{99}$, $H_s$, $T_s$ and $P$ along the GC, with the exception of the multi-year mean of $H_{99}$, which is practically the same throughout the Gulf (see Figure 5a). The multi-year mean values of $H_s$, $T_s$ and $P$ gradually decreased northward (see Figure 5b–d). This is consistent with the spatial variability of the mean wind intensity, which also gradually decreased to the north, as observed by [47] using QuikScat wind data. Moreover, the long-term trends of $H_{99}$, $H_s$ and $P$ increased in all the Gulf (see Figure 6a,b,d), whereas the long-term trend of $T_s$ decreased in both the lower and middle Gulf and increased in the upper Gulf (Figure 6c). The rate of increase was greater for $H_{99}$ (Figure 6a) compared to $H_s$ (Figure 6b), which is also consistent with the global trend of greater rates for extreme events compared to mean conditions [13]. According to [48], an increase in wave heights is primarily driven by increases in surface wind energy, so the long-term upward trend of $H_{99}$ and $H_s$ is thus probably related to a long-term upward trend in wind intensity. Contrary to the general idea that not only the wave heights but also the wave periods increase, this study found a long-term downward trend for $T_s$ in the lower and middle parts of the Gulf. This issue needs further investigation to elucidate the causes and consequences of steeper waves. The long-term trend of $P$ indicated that waves were getting more energetic despite the downward trend given by $T_s$. In agreement with [1], this study shows that $P$ is a better indicator of the long-term behavior of wave

climate than other parameters, such as the wave height taken alone, because *P* includes information on wave heights and periods (see Equation (2)) and can also represent the accumulated wave energy over periods of time. The results of this study suggest that climate change is modifying the long-term climate waves in mean and extreme conditions along the GC, where the trends are significant and comparable with the trends of other closed or semi-enclosed basins around the globe (i.e., Red Sea [49], Black Sea [50] and Persian Gulf [46]).

On the other hand, the long-term shoreline analysis revealed that 2283.6 km (67.8% of the total coast) of the coastline is suffering erosion at a mean rate of −4.2 m per year, while the remaining 1087.4 km (32.2% of the total coast) is under accretion at a mean rate of 1.9 m per year. The most eroded coastal strips are located on the northern and southern coasts of Sonora, on the northern coast of B.C., and along the coast of Sinaloa, whereas the most accretional coastal strips are located along the coasts of B.C.S. and southern B.C.

The distribution of the erosion and accretion points along the GC coast and the dominance of erosion processes can be explained by natural processes, human activities and climate change. Figure 7 shows the tracks of tropical cyclones (hurricanes and depressions) that passed through the GC region between 1980 and 2019. In this period, a total of 63 cyclones occurred in the GC region, 44 of which made landfall at the GC coast, i.e., 1.1 cyclones per year on average impacted the GC coast. Of the 44 tropical cyclones that made landfall at the GC coast, 11 did so in more than one state. The numbers of times tropical cyclones made landfall in the GC region are shown in Table 4: Sinaloa and B.C.S. are the states hit by the highest numbers of tropical cyclones. This observation coincides with [51], who defined these two states as tropical cyclone hotspots for the period 1970–2010. Table 5 shows the monthly average number of cold fronts that passed through Mexico from 1981 to 2019. In this period, 50 cold fronts per year, on average, occurred in Mexico. Cold fronts usually enter Mexico from the central, southern and southeastern parts of the United States of America, and occasionally from the north west of Mexico, the latter situation being more intense [52] and the most recurrent one for the GC.

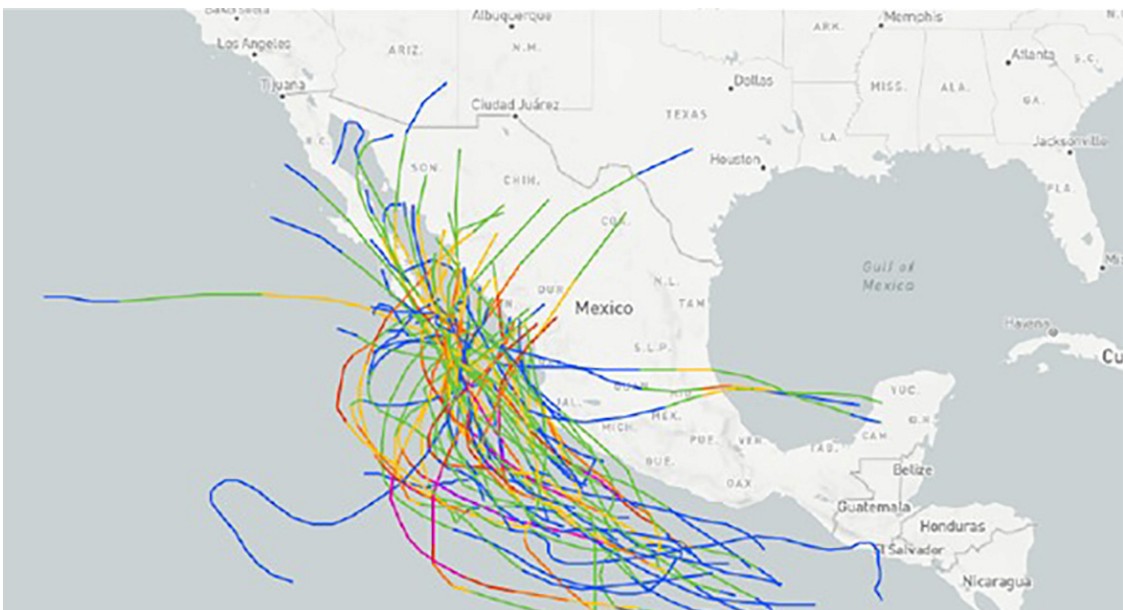

**Figure 7.** Tracks of tropical storms that passed along the GC region from 1980 to 2019. Source: National Oceanic and Atmospheric Administration (2020).

**Table 4.** Number of tropical depressions (td), tropical storms (ts) and hurricanes (h) category 1 to 5 making landfall in the GC during the period 1980–2019. Source: National Oceanic and Atmospheric Administration (2020).

| Landfall Location | Type of Tropical Cyclone | | | | | | | |
|---|---|---|---|---|---|---|---|---|
| | td | ts | h1 | h2 | h3 | h4 | h5 | Total |
| B.C. | 2 | 1 | — | — | — | — | — | 3 |
| B.C.S. | 4 | 7 | 5 | 3 | 2 | — | — | 21 |
| Sonora | 3 | 2 | 2 | — | — | — | — | 7 |
| Sinaloa | 5 | 6 | 5 | 5 | 3 | — | — | 24 |
| total | 14 | 16 | 12 | 8 | 5 | — | — | 55 |

**Table 5.** Monthly average of cold front occurrence in Mexico between 1981 and 2019. Source: National Meteorological Service of Mexico (2020).

| N° | Month | Cold Fronts |
|---|---|---|
| 1 | September | 3 |
| 2 | October | 5 |
| 3 | November | 6 |
| 4 | December | 6 |
| 5 | January | 7 |
| 6 | February | 8 |
| 7 | March | 6 |
| 8 | April | 5 |
| 9 | May | 4 |
| | Total | 50 |

Storm waves and storm surges, associated with tropical cyclones and cold fronts, can induce a number of potential hazards such as beach and dune erosion, overwash processes, inundation and coastal facility damage [53,54]. Figure 8 shows the episodic impact of tropical cyclones and cumulative impact of cold fronts on the GC coast. Although the impact of such extreme events on the GC coast is real and has been highlighted, its study has received little attention. These weather systems have been studied more from the meteorological and climatological point of view than through a coastal geomorphology approach.

Human activities such as the construction of dams, the high population density, and the main economic activities including tourism, agriculture and aquaculture on coastal areas have caused or accelerated the process of erosion or accretion of the GC coastline [55–57]. This anthropogenic pressure on the coast is modifying the sedimentary dynamics, which ultimately affects the shoreline and can still be felt after several years [58–61].

Recent investigations on CC in the region suggest an ostensible rise in the sea level and storm surge elevations [56]. The sea level rise in the Gulf appears to be lower compared to the global mean sea level rise; however, studies suggest that even moderate rates of sea level rise can cause significant shoreline retreat [62]. This shoreline retreat is due to permanent passive submersion (which may affect flat and low-lying areas) or coastal erosion (caused by coastal sediment redistribution due to waves and currents and their interactions with human intervention) [62]. In addition to the sea level, the wave energy has increased (as found in the present work), which may be the cause of the long-term shoreline change presented here, mainly on the east coast, where the relatively lower relief and higher human activity, in comparison to the west coast, suggest that it is more vulnerable to the effects of CC.

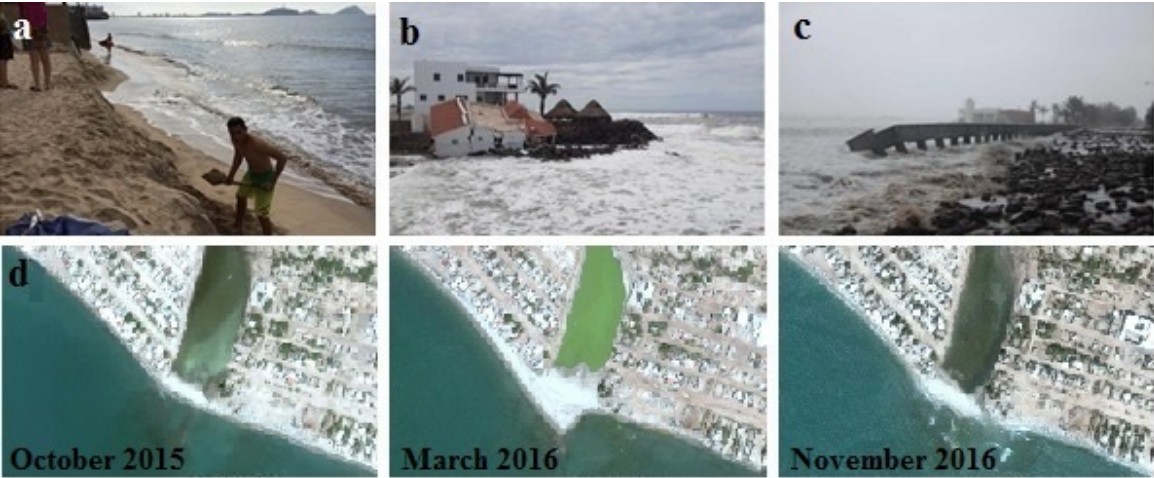

**Figure 8.** Example of the impacts of tropical cyclones and cold fronts on the GC coast: (**a**) Mazatlán beach in Sinaloa, affected by cyclone Willa in 2018; (**b**) Altata beach in Sinaloa, affected by cyclone Bud in 2018; (**c**) Las Glorias beach in Sinaloa, affected by cyclone Odile in 2014; (**d**) Las Bocas beach in Sonora, affected by cold front passages between October 2015 and March 2016. Source: Google Earth and newspapers Noroeste, El Universal and La Pared.

Discerning between changes induced by natural processes, human activities and climate change in the Gulf coast, like in other parts of the world, is a very complicated task, because these factors are interrelated in time and space [63]. Nevertheless, natural processes and human activities in coastal areas are the main factors linked to the modern dynamics [64]. Instead, the shoreline dynamics on sandy coasts will be essentially governed in the future by CC-driven variations in mean sea levels, wave conditions, storm surges and river flows [2].

It is necessary to continue the analysis of the shoreline variability with the purpose of obtaining detailed information about the changes produced by seasonal variation, tropical cyclones and cold fronts. The gaps in understanding the relative influential role of coastal geomorphology, hydrodynamic processes, sediment availability, human intervention and CC highlight the requirement for further investigation that is reliant on observational and modelled data at a local scale. Additionally, research about the influence of river-damming processes on sediment arriving at coastal areas is an important topic for further work considering the spatial relationship among coastline segments with erosion and the presence of dammed rivers. As widely suggested, the authorities should consider the climatic change impacts in the policy framework related to shoreline and coastal management, and the construction of human infrastructure.

## 6. Conclusions

Understanding the long-term variability of the wave period, wave height, wave power and shoreline change will contribute to more reliable assessments of CC's impact on the coast. To date, a regional and long-term perspective study of the wave and shoreline dynamics in the GC was not available, which limited the effectiveness of adaptation strategies. These variabilities were analyzed in the present work using time series of wave data from the ERA5 dataset and historical Landsat images from the USGS.

The results indicate that mean and extreme waves have been becoming more energetic over the years in the GC. The largest increasing trend in wave energy was observed at the middle part of the Gulf, followed by the upper Gulf and, finally, the lower Gulf. The wave height was found to increase in the entire Gulf, while the long-term trend of the wave period was seen to decrease in the lower and middle Gulf and, in turn, was shown to increase in the upper Gulf. This discrepancy calls for further research including in situ measurements. Regarding the long-term coastline change, two-thirds of the

GC coast was found to suffer erosion. Erosional hotspots were identified on the coastal fringes on the northern and southern coasts of Sonora, on the northern coast of B.C. and along the coast of Sinaloa. Intervention should be then prioritized in those coastal fringes. Although this paper did not examine the exact evidence for coastal forcing as a driver of the observed shoreline displacements, a literature review allows arguing that they are a response to the combination of the effects of natural processes with those of human activities and climate change.

The results of this paper present clear relevance for coastal risk management policies and should be incorporated into coastal climate change impact assessments for the development of local-level adaptation plans in the GC. Furthermore, additional research on other trends in oceanic forces that were not addressed in this paper such as storm waves and storm surges is encouraged.

This paper demonstrates that a long-term analysis of wave climate and shoreline change using Interim ERA5 reanalysis wave data and historical Landsat images can be a reliable approach for a coastal environment with limited observational data.

**Author Contributions:** Conceptualization, C.F.-O., Y.Z.-M., M.E. and E.M.; methodology, C.F.-O., Y.Z.-M., W.P.-R., S.M.-A. and Y.R.-C.; formal analysis, C.F.-O., Y.Z.-M., W.P.-R. and S.M.-A.; investigation, C.F.-O., Y.Z.-M., W.P.-R. and S.M.-A.; writing—original draft preparation, C.F.-O., Y.R.-C., M.E. and E.M.; writing—review and editing, C.F.-O., Y.R.-C., M.E. and E.M. All authors have read and agreed to the published version of the manuscript.

**Funding:** This research was funded by the Autonomous University of Sinaloa under research grant number PROFAPI2014/073.

**Acknowledgments:** The authors are very grateful to Sébastien de BRYE for his helpful and unselfish support in reviewing this paper.

**Conflicts of Interest:** The authors declare no conflict of interest. The funders had no role in the design of the study; in the collection, analyses, or interpretation of data; in the writing of the manuscript; or in the decision to publish the results.

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
