# Peer review of "Long-Term Analysis of Wave Climate and Shoreline Change along the Gulf of California"

_applsci, doi:10.3390/app10238719_

Round 1

Reviewer 1 Report

Paper title: Long term analysis of wave climate and shoreline change along the Gulf of California

Authors: Cuauhtémoc Franco-Ochoa, Yedid Zambrano-Medina , Wenseslao Plata-Rocha , Sergio Monjardín-Armenta , Yandy Rodríguez-Cueto , Mireille Escudero , Edgar Mendoza

Journal: Applied Science MDPI

General comment

The paper deals with the effects of the climate change on the shoreline retreat with specific attention to the Gulf of California. In particular the paper investigates the trend of wave periods, wave heights and power using ERA5 database for 40 years of data, and relates this trend with historical Landsat images taken for 40 years. The results show that there is an increasing trend over the years  with clear implications for coastal risk management.

The paper is well written and suggests interesting applications in term of risk protection. It deserves publication on Applied Science once some minor points have been fixed according to the specific comments reported below.

Specific points

Line 23: has should be replaced with have

Line 55: 0.8 is it correct? The uncertainty is equal to the target value

Line 104: there should be removed

Line 126: October should be changed in October and November

Table 1: there are some date errors (1978-11-78; 2019-11-31, etc.)

Line 140: (NIR): move the explanation (near infrared) from line 146.

Line 140-141: some acronyms should be specified: MSS, OLI.

Line 146: same as before: NDWI

Line 158 and following: some details on the adopted software and on the used algorithms should be provided.

Line 160-161: not clear the explanation of EPR, which years, which dates?

Tab. 4: can be synthesized, reporting only the number of hurricanes per level and per area

Author Response

Thank you for the comments on our manuscript entitled "Long-term analysis of wave climate and shoreline change along the Gulf of California" (applsci-1020225.). We appreciate the suggested modifications and have revised the manuscript accordingly. The detailed responses to the reviewers’ comments are shown in red text and presented as follows:

Reviewer 1

*General comment

The paper deals with the effects of the climate change on the shoreline retreat with specific attention to the Gulf of California. In particular the paper investigates the trend of wave periods, wave heights and power using ERA5 database for 40 years of data, and relates this trend with historical Landsat images taken for 40 years. The results show that there is an increasing trend over the years with clear implications for coastal risk management.

The paper is well written and suggests interesting applications in term of risk protection. It deserves publication on Applied Science once some minor points have been fixed according to the specific comments reported below.

*Specific points

Line 23: has should be replaced with have

Response: We replaced “has” with “have” on Line 23.

…….no previous studies have tackled the long-term trend of wave climate at a regional scale.

Line 55: 0.8 is it correct? The uncertainty is equal to the target value

Response: 0.8 ± 0.8 mm per year is correct. This information was taken from Páez-Osuna, F.; Sanchez-Cabeza, J.A.; Ruiz-Fernández, A.C.; R. Alonso-Rodríguez, A.C.; Piñón-Gimate, A.; Cardoso-Mohedano, J.G.; Flores-Verdugo, F.J.; Carballo, J.L.; Cisneros-Mata, M.A.; Álvarez-Borrego, S. Environmental status of the Gulf of California: A review of responses to climate change and climate variability. Earth-Science Rev. 2016, 162, 253–268.

Line 104: there should be removed

Response: Line 104 indicates “medium in size (0.25–0.50 mm) and the mean beach slope is 4°. The river network there is almost” …. We modified Line 103 to clarify that we were referring to the east coast.

On the east coast, beach sands are mainly medium in size (0.25–0.50 mm) and the mean beach slope is 4°.

Line 126: October should be changed in October and November

Response: We replaced "October" with "autumn" because the images were also taken in late September and during November.

All the images were collected in autumn in order to reduce the …….

Table 1: there are some date errors (1978-11-78; 2019-11-31, etc.)

Response: We replaced the following dates from Table 1:

“1978-11-78” with “1978-11-08”

“2019-11-31” with “2019-10-31”

“2019-11-22” with “2019-10-22”

“2019-11-29” with “2019-10-29”

“2019-11-04” with “2019-10-04”

Line 140: (NIR): move the explanation (near infrared) from line 146.

Response: The following text was modified:

After that, the Normalized Difference Water Index (NDWI) [1] was calculated with bands 4 (Green) and 6 (Near InfraRed [NIR]) of the MSS sensor, and bands 3 (Green) and 5 (NIR) of the OLI sensor,…..

Line 140-141: some acronyms should be specified: MSS, OLI.

Response: The acronyms MSS and OLI was specified in Table 1.

Multispectral Scanner System (MSS) and Operational Land Imager (OLI).

Line 146: same as before: NDWI

Response: The acronym NDWI was specified in Line 137.

…… Normalized Difference Water Index (NDWI) [35].

Line 158 and following: some details on the adopted software and on the used algorithms should be provided.

Response: The following text was modified:

Shoreline changes were analyzed using DSAS (version 5.0), which is a software extension for ArcGIS©, developed by USGS to calculate shoreline change rates from multiple historic shoreline positions [19]. To use DSAS, a geographic database was created that combined digitized shorelines and a baseline in front of the coast, then cross-shore transects every 100 m were generated along the west and east coasts of the GC.

Line 160-161: not clear the explanation of EPR, which years, which dates?

Response: We defined the EPR in a general way as:

EPR is the division of the distance between the shoreline position at different dates by the difference in years of the shoreline position.

The dates of the shoreline used in our study for the calculation of the EPR are indicated in Figure 3.

Tab. 4: can be synthesized, reporting only the number of hurricanes per level and per area

Response: We synthesized the Table 4 as follows:

Table 4. Number of tropical depressions (td), tropical storm (ts) and hurricanes (h) category 1 to 5 making landfall in the GC during the period 1980–2019. Source: National Oceanic and Atmospheric Administration (2020).

Landfall location

Type of tropical cyclone

td

ts

h1

h2

h3

h4

h5

total

B.C.

2

1

---

---

---

---

---

3

B.C.S.

4

7

5

3

2

---

---

21

Sonora

3

2

2

---

---

---

---

7

Sinaloa

5

6

5

5

3

---

---

24

total

14

16

12

8

5

---

---

55

Author Response

Reviewer 2

This manuscript provide results of long-term changes of shoreline positions and wave conditions along the coasts of GC using ERA 5 reanalysis data and Landsat images. The provided information may be important to understand the coastal processes caused by human activities and climate change in this region. However, the manuscript is poorly organized and I do not recommend this paper to be published in the present form.

Response: Thank you for the time devoted reading our manuscript and for all this comment. We have reviewed all the work and made some changes that may help the readers go through it.

1) The shoreline positions were estimated by finding the boundaries between water and land based on NDWI. However, I do not understand how these boundaries can be used to detect the HWL where the maximum tidal range reaches up to ~3.5m in the upper Gulf? The possible error range has to be considered before making the results on the shoreline position changes.

In addition, there could be wave breaking zones (appear as white bubbles) in the satellite images. Please explain how there breaking zones were treated when detecting the boundaries between the land and water.    

Response: During the digitalization process we didn’t got the water-land boundary but the HWL seen in the image as the upper limit of wet sand. We have modified the text to clear this point as follows:

3.2. Image processing

Two major image processing operations were carried out in this study. First, radiometric and atmospheric correction was applied to all satellite images of Table 1 in the ENVI 5.3 software [33]. The Reflectance Radiometric Calibration method was used to sharpen the satellite images, while the Fast Line of Sight Atmospheric Analysis of Spectral Hypercubes (FLAASH) method eliminated disturbances caused by the atmosphere [34]. After that, the Normalized Difference Water Index (NDWI) [35] was calculated with bands 4 (Green) and 6 (Near InfraRed [NIR]) of the MSS sensor, and bands 3 (Green) and 5 (NIR) of the OLI sensor, through Equation :

(1)

The relation between the green and NIR bands highlights in the images the conditions of high soil moisture and water masses [35-37]. In turn, it highlights the upper wetting limit of the foreshore caused by the highest run-up of the last high tide before the satellite image acquisition date.

It is also worth mentioning that in the manuscript we indicated the following

“The High Water Line (HWL) is widely used in the investigation of shoreline change to represent the shoreline [38]. It is defined as the highest run-up of the last high tide [39] and visually discernible in satellite image [16]. Its demarcation in the images is difficult due to the lack of knowledge of the oceanographic conditions at the time of the aerial view. Nevertheless, it has been shown that the wet/dry line (on a rising tide is equal to maximum run-up limit; on a falling tide is equal to part of beach that is still wet [16]) closely approximates the HWL [40,41]. For these reasons, in this paper, the wet/dry line from Landsat images was used as an approximation of the HWL inland/outland horizontal displacement.”

The use of the wet/dry line as an approximation of the HWL reduces the error due to the variation of the tide. In addition, the objective of the shoreline analysis was to evaluate the erosion and accretion long trends and not the precise quantification of the shoreline displacements. The accuracy of the results was sufficient to identify the Erosional hotspots.

With regard to how there breaking zones were treated when detecting the boundaries between the land and water, the NDWI is designed to compute the water content on land cover due to the index use NIR or SWIR, depending on the index version, bands that do not reflect any energy in water bodies. Breaking zone has a water content equal to steady-state water surface [1]. In turn, detecting the wet sand upper level, minimizes the effect of wave braking in the digitization process.

  1. McFeeters, S.K. The use of the Normalized Difference Water Index (NDWI) in the delineation of open water features. Int. J. Remote Sens. 1996, 17, 1425–1432.

The following text was modified:

Lines 151 to 153: For these reasons, in this paper, the wet/dry line from Landsat images was used as an approximation of the HWL inland/outland horizontal displacement.

Lines 154 to line 156: Given the latter and the resolution of images, the objective of the analysis was to evaluate the erosion and accretion long trends and not the precise quantification of the shoreline displacements based on the position of HWL.

2) One of the main findings in this study is that the erosion in GC has become more serious due to the increased wave power during the period. However, it seems to me that the erosion/accretion pattern in Figure 4 shows high locality rather than general pattern. If the erosion has increased due to increasing wave energy, then the erosion pattern has to be more general along the coast. In addition, the east coast of GC (Figure 4(c) & (d)) has erosions while accretions generally occur in the west coast (Figure 4(a) & (b)), which may not be caused by the changes on the wave conditions in general (or the wave conditions were different at each side of the Gulf).

In my opinion, the high locality of the erosion pattern may be due to the reduced sediment input by the dammed rivers or by other reasons to cause the erosion in the locations.

My suggestion is that the authors should be more specific in analyzing their data to support their arguments.          

Response: We agree that the erosion/accretion pattern in Figure 4 shows local effects, this would be expected in such a long coastline. The local response is due to several issues such as the presence of islands, particularities of the bathymetry and geological aspects of the coast. Our paper did not examine the exact evidence of coastal forcing as a driver of the observed shoreline displacements, nevertheless, a literature review allows to argue that they are a response to the combination of the effects of natural processes with those of human activities and climate change. We indicated that discerning between changes by natural processes, human activities and climate change in the Gulf of California coast, like in other parts of the world, is a very complicated task, because these factors are interrelated in time and space [2]. Nevertheless, natural processes and human activities in coastal areas are the main factors linked to the modern dynamics [3]. Instead, in the future the shoreline dynamics on sandy coasts (like the major part of the coastline in the Gulf of California) will be essentially governed by climate change driven variations in mean sea level, wave conditions, storm surges, and river flows [4].

  1. Gómez-Pazo, A.; Pérez-Alberti, A.; Pérez, X.L.O. Recent evolution (1956-2017) of rodas beach on the Cíes Islands, Galicia, NW Spain. J. Mar. Sci. Eng. 2019, 7.
  2. Bera, R.; Maiti, R. Quantitative analysis of erosion and accretion (1975–2017) using DSAS — A study on Indian Sundarbans. Reg. Stud. Mar. Sci. 2019, 28, 100583.
  3. Ranasinghe, R. Assessing climate change impacts on open sandy coasts: A review. Earth-Science Rev. 2016, 160, 320–332.

Figure 4 helped identifying the erosional hotspots where intervention can be prioritized and the results from the long-term wave climate analysis can be integrated into coastal climate change impact assessments for the development of local level adaptation plans in the Gulf of California. This will encourage further investigation that is reliant on observational and modelled data at a local scale, which will contribute to a better understanding of the relative role of local characteristics (coastal geomorphology, hydrodynamic processes, sediment availability, human intervention and climate change) in coastal response.

The following text was added:

Lines 342 and 344: Additionally, the research about the influence of river damming processes on sediment arriving to coastal areas is an important topic for further works considering the spatial relationship among coastline segments with erosion and presence of dammed rivers.

We hope that our manuscript will achieve your approval. If not, we are available to resolve any issue or proceed with further revisions as necessary.

Thank you!

Round 2

Reviewer 2 Report

The manuscript has been properly revised.